# Precision Tools in Immuno-Oncology: Synthetic Gene Circuits for Cancer Immunotherapy

**DOI:** 10.3390/vaccines8040732

**Published:** 2020-12-03

**Authors:** Giuliano Bonfá, Juan Blazquez-Roman, Rita Tarnai, Velia Siciliano

**Affiliations:** Synthetic and Systems Biology Lab for Biomedicine, Istituto Italiano di Tecnologia-IIT, Largo Barsanti e Matteucci, 80125 Naples, Italy; giuliano.bonfa@iit.it (G.B.); juan.blazquez@iit.it (J.B.-R.); rita.tarnai@iit.it (R.T.)

**Keywords:** synthetic biology, synthetic immunology, cell-based therapies, T-cell engineering, T-cell immunotherapies, Boolean logic in T cells

## Abstract

Engineered mammalian cells for medical purposes are becoming a clinically relevant reality thanks to advances in synthetic biology that allow enhanced reliability and safety of cell-based therapies. However, their application is still hampered by challenges including time-consuming design-and-test cycle iterations and costs. For example, in the field of cancer immunotherapy, CAR-T cells targeting CD19 have already been clinically approved to treat several types of leukemia, but their use in the context of solid tumors is still quite inefficient, with additional issues related to the adequate quality control for clinical use. These limitations can be overtaken by innovative bioengineering approaches currently in development. Here we present an overview of recent synthetic biology strategies for mammalian cell therapies, with a special focus on the genetic engineering improvements on CAR-T cells, discussing scenarios for the next generation of genetic circuits for cancer immunotherapy.

## 1. The Essential of Cancer Immunotherapies

Cancer immunotherapy has the goal of improving anti-tumor immune responses reducing off-target effects typical of chemotherapies and other state-of-the-art treatments. Since cancer cells often evade the surveillance of the immune system, immunotherapies have the function of priming the immune response to make it more efficient. Different classes of immunotherapies have already been approved for cancer treatment and some others are in clinical trials [1] intending to facilitate the recognition of cancer cells by the immune system [2,3]. These include checkpoint inhibitors, lymphocyte activating cytokines, agonists for co-stimulatory receptors, cancer vaccines, oncolytic viruses, bispecific antibodies and T cell-based adoptive immunotherapy (ACT) [2,3]. Checkpoint inhibitors are the most relevant and largely studied immunotherapeutic drugs up to date. They act by blocking co-inhibitory molecules binding to their cognate ligands on the surface of cancer cells. The two most common strategies are the PD-1/PD-L1 axis blockade and the inhibition of CTLA-4, to prolong T cell activity and anti-tumoral effects [4]. A limitation of immune checkpoint inhibitors is that they can cause immune-related adverse events (irAEs), usually related to autoimmunity in a dose-independent manner [5,6]. In addition, many patients do not respond to this therapy due to the low number of tumor-infiltrating lymphocytes (TILs), downregulation of co-inhibitory molecules in both tumor cells and T cells, and adapted resistance [4,7].

Cytokines act by actively limiting tumor growth with a direct anti-proliferative or pro-apoptotic action or by enhancing tumor recognition and cytotoxicity of the immune system against cancer cells [8]. Nevertheless, several issues including the pleiotropic function of many cytokines, which can act both as immunosuppressors or activators depending on the cellular context, the redundancy of cytokine signaling, and the short half-life of these molecules make the efficacy of these treatments difficult. This therapy often consists of the administration of high doses of cytokines that can cause vascular leakage and cytokine release syndrome (CRS) [9]. A recent strategy, still at an earlier phase of development, is the use of agonist antibodies designed to specifically bind co-stimulatory molecules, such as CD28 or the inducible T cell co-stimulator (ICOS), increasing T cell proliferation and survival in the immunoglobulin-like superfamily, as well as OX40, CD27, and 4-1BB in the TNFR superfamily [10]. Of note, agonist antibodies are designed to bind and activate the target receptor, mimicking the action of the natural ligand, and the kinetics of this binding may depend on many factors, such as the affinity to the epitope, and the interaction between the antibody fragment crystallizable domain (Fc) and its receptor. 

Lastly, cancer vaccines represent another promising strategy to increase the immune response against cancer. They are divided into (i) cell vaccines, obtained from the tumor or immune cells, (ii) protein/peptides vaccines based on tumor-associated antigens (TAAs), and (iii) genetic vaccines, which use viral, plasmid vectors, or mRNA from autologous tumor tissue to deliver the antigen. 

Cancer vaccines can be either prophylactic or therapeutic. Examples of prophylactic vaccines include the one against hepatitis B virus which accounts for liver cancer, and the one against human papillomavirus, responsible for the most cervical cancers [11]. 

Therapeutic vaccines have the goal to increase the tumor-specific adaptive response. Cancer vaccines have shown reduced toxicity and autoimmunity issues, although their efficacy is, for now, lower than other immunotherapeutic strategies, due to (i) an inappropriate activation of effector cells, (ii) limited accumulation of these cells in the tumor, and (iii) to the immunosuppressive tumor microenvironment. Advances in synthetic biology allowed engineering safer and more effective vectors, DNA or RNA–based implemented on bacterial or viral backbones, carrying genetic components, and functional resources [12]. 

## 2. T Cell-Based Therapies

T cells have a prominent key role in the immune response against cancer. TILs are commonly found in the tumor microenvironment (TME), where they can exert anti-tumor actions [13]. Their presence is frequently associated with a better prognosis, even though a cytotoxic subset such as CD8^+^ T cells may undergo impaired activation in the TME. The isolation of these cells from excised tumors and the subsequent reinvigoration in vitro before reintroducing them back to the patient is one of the strategies currently used in clinics to increase T cells response against tumor [13]. 

Leveraging on immune cell infiltration in the TME, over the past years, several therapies have been developed to improve their action against tumors. In particular, T cell-based adoptive immunotherapy (ACT) is a novel anticancer treatment that consists of the in vitro expansion and activation of autologous immune cells, prior to reinfusion into patients [14]. So far, this treatment has been used mainly for hematologic cancers, although they have been recently tested also for solid tumors [15]. ACTs include (i) tumor-infiltrating lymphocytes (TILs) from an endogenous source, (ii) T cell receptor (TCR)-modified T cells, and (iii) chimeric antigen receptor T cells (CAR-T cell) [16]. The use of TILs for cancer treatment is achieved by harvesting CD4^+^ and CD8^+^ T lymphocytes from the patient’s tumor, followed by in vitro expansion in a medium supplemented with IL-2 alone or in combination with IL-7, IL-15, and IL-21. TILs are composed of an enriched polyclonal population and are reinfused in the patient to increase the immune response by recognizing tumor-associated antigens as well as tumor neoantigens, and mutated antigens expressed only by tumor cells [17].

TCR-modified T cells are first isolated from tumor patients, and genetically modified with a TCR that is engineered to specifically recognize antigens expressed on cancer cells presented by the MHC system [18,19]. Instead, CAR-T cells are genetically engineered to express a Chimeric Antigen Receptor (CAR) composed of two domains: (1) an extracellular domain consisting of an antibody single-chain fragment (ScFv) that specifically recognizes cell surface antigens on tumor cells and (2) a chimeric intracellular domain formed by the activating intracellular domains from TCR complex (CD3ζ) and other co-stimulatory molecules. Unlike conventional TCR-mediated activation, CAR-mediated activation is antigen-specific and MHC-unrestricted [20,21]. CAR-T has proven higher efficacy, at least against hematological malignancies such as B cell neoplasia and multiple myeloma.

Unfortunately, CAR-T therapy does not exhibit the same efficacy in solid tumors as it does for blood cancers. This is due to several reasons among which tumor heterogeneity and T cell dysfunction are caused by tonic signaling coming from tumor cells and chronic antigen exposure. Additionally, the immunosuppressive features of the tumor microenvironment (i.e., hypoxia), the presence of inhibitory myeloid-derived cells (neutrophils, M2 macrophages, myeloid-derived suppressor cells) and regulatory T cells (Tregs), and the inhibitory role of the extracellular matrix, composed by fibrous proteins, collagen, and hyaluronan [22], have been described as potential immune escape mechanisms of solid tumors in the scientific literature. Moreover, solid tumors induce the formation of aberrant tumor vasculature by producing molecules such as VEGF and other proangiogenic factors, which induce the expression of inhibitory receptors, like PD-1, TIM-3, and IDO-1 on relevant cytotoxic immune cell subsets [23].

All these different features have contributed to classifying tumors in cold and hot tumors depending on the amount and relative quantities of lymphocytes at the tumor core and the tumor margin [24]. Hot tumors, that have a high level of infiltrating lymphocytes, represent the best candidates for immune checkpoint inhibition therapies or cell therapies, whereas cold tumors display the lowest response rate [24]. Several approaches have been proposed to increase the response of those tumors to immunotherapy. Among them, combination therapies aiming to enhance T cell responses by removing co-inhibitory signals, such as immune checkpoint inhibitors (ICIs) and myeloid-derived suppressor cells (MDSC) depletion, along with the activation of co-stimulatory signals such as anti-OX40, are currently being developed. Unfortunately, autoimmunity-related issues represent a major limitation when these approaches are implemented in patients [25]. 

It has been reported that CAR-T therapy exhibits some side effects, including neurotoxicity and Cytokine Release Syndrome (CRS), that to date are not deeply understood. Symptoms of neurotoxicity include among others headache, confusion, aphasia, attention deficit, memory loss, and only in severe cases cerebral edema, that may lead to death. In the vast majority of the cases, their onset is 4–5 days post-infusion, but are usually reversible and solved in 3–8 weeks after CAR-T cell injection. CRS usually occurs in up to 22 days post-injection and it is solved in 60 days. Symptoms include fever, nausea, fatigue, hypotension, and hypoxia. The involved cytokines in CRS following CAR-T cell therapy includes not only effector cytokines such as interferon (IFN)-γ, IL-2, IL-6, and granulocyte-macrophage colony-stimulating factor (GM-CSF) but also the cytokines mainly secreted by the monocytes and/or macrophages such as IL-1, IL-6, IL-8, IL-10, IL-12, tumor necrosis factor (TNF)-α and IFN-α. Neurotoxicity and CRS may be considered linked since they are derived from an extreme immune activation due both to CAR-T cells and non-CAR-T cells [26,27].

## 3. Synthetic Biology for Immunotherapy

Synthetic biology has become an innovative solution for the design of new, effective strategies in many fields including cell-based immunotherapies [28]. The use of genetic circuits for smart cell therapies stems from the concept of sensing soluble or membrane-associated disease markers and respond with a therapeutic output. Over the past two decades, synthetic biology flourished to provide solutions for tight and reliable characterization of synthetic modules both in cell-free systems [29,30] and in cellular testbeds [31], aided by mathematical models [32] and computational design [33]. In the following paragraphs, we will explore synthetic biology strategies applied to mammalian cell therapy and cancer immunotherapy.

### Synthetic Biology Approaches for Mammalian Cells Therapy and Cancer Immunotherapy

Synthetic circuits take advantage of the capability of the cell to sense extracellular or intracellular stimuli by several means [34]. For example, pathways from natural cell receptors that sense the extracellular environment are rewired to drive circuit activation and therapeutic response. In this regard, G protein-coupled receptors (GPCRs) and ion channels have been ectopically expressed in mammalian cells to sense disease markers (e.g., glucose and dopamine levels, bile acid, fatty acid, and formyl peptides) and express a therapeutic protein [35]. A distinct feature of synthetic circuits is the design of Boolean logic gates, which work analogously to electronic systems. These logic gates imply the capacity of the genetic device to compute the information deriving from multiple inputs to trigger the expression of the desired output in a tightly specific fashion (explored in more detail in topic 4.3 below). For example, an “AND” logic gate indicates that two or more inputs must co-exist to activate the output.

Cytokine receptors have been exploited to construct such logic gates, like in the case of an “AND” gate that uses TNF (tumor necrosis factor) and IL-22 (interleukin 22) receptors to sense psoriasis and to drive anti-inflammatory IL-4 and IL-10 in response [36]. Thus, the presence of both TNF and IL-22 will trigger the desired response. Toll-like receptors (TLRs) such as TLR2 and TLR1/6 have also been expressed in HEK-293 cells (in association with CD14) to sense the presence of *Staphylococcus aureus*, coupled to a rewired downstream signaling pathway to express a bacteriolytic lethal enzyme and cure MRSA (methicillin-resistant *S. aureus*) [37].

Although the use of natural receptors is facilitated and the performance control of components are tuned through evolution, these approaches may suffer crosstalk with other native signaling pathways and lack of modularity. Differently, synthetic receptor systems can couple sensing of soluble molecules to engineered functions in a more precise/modular approach. For example, GEMS (generalized extracellular molecules sensors) relies on native signaling pathways and are modular [38]. This approach is composed of a scaffold molecule, the Epo receptor (EpoR) fused at the N-terminus to extracellular domains that enable binding to different targets, and at the C-terminus to intracellular domains that dimerize upon binding of input molecules activating different signaling pathways. By co-transfecting a synthetic output expression cassette that responds to transcriptional activation of each signaling pathway, various therapeutic proteins can be expressed in response to input proteins such as disease markers. The system was demonstrated for the activation of JAK/STAT (Janus kinase/signal transducer and activator of transcription), MAPK (mitogen-activated protein kinase), PLCγ (phospholipase C gamma) or PI3K/Akt (phosphatidylinositol 3-kinase/protein kinase B; induced by VEGFR2) signaling pathways, connected downstream to responsive synthetic promoters. Another modular system with a great therapeutic potential called GEARs (Generalized Engineered Activation Regulators) was recently created for the direct repurposing of an endogenous signaling pathway to activate native or synthetic promoters [39]. In this system, natural or chimeric receptors sense a soluble stimulus that is transmitted to an intracellular natural regulatory or transactivation domain of key signaling pathways. The last one is fused to MS2 bacteriophage coat protein (MCP) that combined with catalytically dead CRISPR associated protein 9 (dCas9) and a synthetic guide RNA (sgRNA) containing two MS2 coat protein-binding loops (MS2) are used to activate desired cellular pathways.

Besides sensing extracellular cues, modular sensing-actuator devices were also created capable to sense intracellular proteins, initiating a programmed transcriptional activation to produce a reporter or therapeutic gene [40,41]. In this approach, the sensing modules are based on intracellular antibodies (intrabodies) designed to recognize medically relevant proteins and the actuator module relies on TEV protease. The TEV-mediated release of a synthetic transcription factor in response to the intracellular proteins drives a chosen output activation. These strategies showed potential therapeutic applicability to treat metabolic and infectious diseases and in the field of cancer immunotherapy to improve CAR-T cells therapy.

In the specific area of cancer immunotherapy, CAR-T therapy represents the state-of-the-art of cell-based therapies. CARs consist of an extracellular scFv domain that recognizes the target antigen, a transmembrane domain, and an intracellular domain to trigger downstream T cell signaling. CAR-T cell design has improved over time leading to several generations that are distinguished for changes in the co-immunostimulatory domains [42]. In particular, third-generation CARs include CD28 or 4-1BB co-stimulatory domains to enhance performance and downstream signaling. Recently, the fourth generation CAR-T associates the expression of an additional effector molecule like IL-2 and IL-12 to fight solid tumors more efficiently. This strategy, called TRUCKs, has the goal of treating cancer patients by stimulating also immune cells other than CAR-T that infiltrate the TME [43,44]. However, it is well known that adoptive cell therapies suffer limitations, and novel bioengineering approaches may help to overcome them. These limitations can be summarized in approaches (1) to improve CAR-T cells treatment efficacy, especially those of solid tumors, and (2) to reduce ON target/OFF tissue toxicities observed in CAR-T cell therapy. A complete list of the main approaches, the engineering rationale, and the purpose is summarized in Table 1 and discussed in detail in the next paragraph.

## 4. Synthetic Biology Approaches to Boost CAR-T Cell Treatment’s Efficacy

### 4.1. Addressing the Tumor Immune Escape

The heterogeneous expression pattern of target antigens and antigen-negative relapses in long-term follow-ups prompted the need of targeting more than one antigen synergistically. An OR-gate CAR can be used to recognize two different tumor-associated antigens, requiring only one of them to activate cells. More recent strategies consist in the integration in the same CAR construct of two scFv domains separated by a protein linker, thus forming a bi-specific CAR namely “Tandem” CAR (Figure 1) [45,46,47]. CD19/CD20, as well as CD19/CD22 bi-specific CAR-T cells, are being currently tested on clinical trials (NCT04007029, NCT04215016, NCT03919526, NCT04303520) for the treatment of B cell malignancies.

An intelligent approach to obtain a universal and modular platform of T cells to bind several different targets is the peptide-specific switchable CAR-T-cells (sCARs). The general idea consists of an extracellular domain of the CAR to recognize a ligand which is provided as a free molecule fused to an scFv specific to the tumor antigen (Figure 1). Thus the ligand-scFv function as a bridge between the “universal” CAR and the tumor cells, and the cytotoxicity is limited only to cells coated with the antibodies/adapters [48,49,50,51]. An elegant evolution of this strategy is the Split Universal Programmable (SUPRA) CARs [52]. Similar to sCARs, SUPRA CARs are universal CARs composed of a leucine zipper that binds a homologous domain fused to an scFv, creating a synapsis between the CAR and the tumor cells (Figure 1). This design allows fine-tuning of T cell activation through the binding affinity of the leucine zippers, by adding competing leucine zippers and playing with the binding affinity of the scFv. 

Overall, these two modular strategies could substantially reduce the costs and time of T cell expansion into antigen-specific CAR-T cells. Furthermore, by adding or withholding the scFv part of the construct it might be able to tightly control CAR-T cells activation, improving CAR-T cells safety.

### 4.2. Rewiring Immunosuppressive Signals from the TME

It is well known that the TME releases several signals which depress the immune response against cancer. For example, TGFβ can induce a shift of cytotoxic T cells towards a Treg-like phenotype with highly immunosuppressive functions in the TME. To counteract this effect, CAR, where the extracellular part of the TGFβRII is fused to the endodomain of 4-1BB or that link a TGFβ-specific scFv to the CD28-CD3ζ intracellular signaling domains, have been engineered (Figure 1), activating rather than inhibiting CAR-T cells by TGFβ stimulation [53,54]. The ability to convert an immunosuppressive stimulus coming from the TME into an immunostimulatory response was achieved also by fusing the IL-4 receptor ectodomain (an immunosuppressive cytokine) to the IL-7 receptor endodomain or the βc receptor subunit common to IL-2 and IL-15 signaling (immunostimulatory cytokines) (Figure 1) [55,56]. Of note, an undesired limitation of these approaches is that rewiring an immunosuppressive stimulus to an activating/co-stimulatory signal might lower T cells activation threshold, thus favoring OFF-target cytotoxicity onset.

Switch receptor combining immune checkpoint proteins ectodomain fused to co-stimulatory proteins endodomains also showed to enhance T cells antitumor immunity. For example, PD-1-CD28 and CTLA-4-CD28 chimeric receptors demonstrated to boost tumor-specific T cells killing of tumor cells [57,58]. Similarly, T cell immunoreceptor with Ig and ITIM domains (TIGIT) modified following the same strategy, showed better control of tumor growth in vivo (Figure 1) [77]. The engagement of endogenous non-engineered T cells by secretion of bispecific T-cell engagers (BiTEs) is a more recent development. In a preliminary approach, these bi-specific antibodies were secreted by engineered cells and consisted of one scFv targeting EGFR, overexpressed by glioblastoma cells, and the scFv on the other side of the molecule targeting CD3. This strategy facilitated the elimination of tumor xenografts in vivo [78].

### 4.3. Boolean Logic Gates Design to Reduce CAR-T Cell Treatments Toxicity

By enhancing multiple signals-derived specificities, Boolean logic gates have been demonstrated to be useful to prevent or reduce toxicity concerns arising from immune cellular therapies. By this approach, CAR-T cells would exert their function only in the presence of the two antigens A and B (AND gate) or if the downstream activation of A is blocked by the presence of inhibitory B (AND NOT gate) (Figure 2) [59,62]. 

Several strategies that implement AND logic gates on engineered T cells have been developed up to date. One model consists of splitting intracellular activation domains into two different CARs recognizing diverse antigens (e.g., CD3ζ signaling domain in one CAR and CD28/4–1BB costimulatory domains in the other, Figure 2A) [59,79]. Similarly, SynNotch CAR-T cells exert their cytotoxic activity only if two different antigens are present on the target cell surface [60]. In this two-steps approach, the binding of the SynNotch receptor to the first TAA triggers the release of an intracellular transcription factor, which drives the expression of a CAR specific to a second TAA. CAR binding to the second TAA will ultimately promote T cell activation against cells expressing both TAA (Figure 2B). Lastly, an AND-like gate is the Cytoplasmic Oncoprotein VErifier and Response Trigger (COVERT) system. Here, T cells bind the TAA to release a conditionally active cytotoxic protein SUMO-GranzymeB (SUMO-GrzB). GrzB is inactive unless SUMO is removed by the SENP1 protease which is selectively overexpressed by tumor cells. Thus, only in cancer cells, the cleavage of SUMO by SENP1 allows GrzB to unleash its cytotoxic capacity [61]. 

To limit OFF-side effects of CAR-T, AND NOT logic gates have also been implemented. The co-expression on CAR-T cells of activating and inactivating CARs (iCAR) have been shown to prevent T cell activation against cells expressing two antigens while killing the ones expressing only the antigen recognized by activating CARs. iCAR design consists of a ligand-binding extracellular domain fused to an immune checkpoint protein endodomain (e.g., PD-1, CTLA-4), thus transmitting inhibitory signals upon antigen encounter, and preventing CAR-T cells activation against cells expressing two TAA (Figure 2C). Specifically, T cells were engineered to co-express a second-generation anti-CD19 CAR along with an iCAR targeting the prostate-specific membrane antigen (PSMA). Engineered iCAR-T cells were effectively activated when in contact with cells expressing only CD19 AND NOT PSMA [62]. In line with this work, Hamburguer et al. recently built-up an optimized iCAR system, namely the Tmod2 system [63]. Here, Tmod2 cells are engineered to co-express an activator CAR or TCR recognizing a TAA expressed by tumor and normal cells, as well as a blocker receptor able to recognize a surface antigen that is lost in tumor cells due to loss of heterozygosity (LOH) (Figure 2D). Thus, engineered cells are tuned ON only if the second antigen is absent from the target cell surface. In this system, signal integration relies on the presence of immunoreceptor tyrosine-based activation motif (ITAM) sequences on activating CARs/TCRs and an immunoreceptor tyrosine-based inhibitory motif (ITIM) segment that mediates blocker receptor inhibitory signaling. This seminal study opens the field of targeting missing antigens, rather than the expressed ones, for cancer immunotherapy.

### 4.4. Strategies to Limit Cytokine-Dependent Off-Target Cytotoxicity

Standard TIL treatment protocols usually require treating patients with repeated high doses of IL-2 which however can also induce the expansion of immunosuppressive Tregs and trigger severe side effects on treated patients [8,80]. Hence, several approaches have been developed to deliver cytokine signaling in engineered T cells, avoiding off-target stimulation (e.g., in Tregs). One of the most original approaches consists of the development of orthogonal IL-2 receptor pairs that are activated by mutated IL-2, which is not able to activate native IL-2 receptors. Thus, with this method, it should be possible to specifically activate engineered T cells avoiding IL-2 OFF-target deleterious effects. This strategy showed to enhance engineered T cells expansion and anti-tumor efficacy, upon exogenous administration of mutant IL-2, in preclinical models [64]. A similar strategy was followed in the engineering of T cells to express constitutively activated IL-7 receptors, thus providing a pro-survival stimulus to engineered cells that leads to enhanced antitumor activity and an increase in T cells persistence in mice [65].

Other approaches trying to confine cytokine signaling to engineered cells involve IL-12 expression. For instance, stably transduced T cells to express IL-12 driven by NFAT synthetic promoter [66] allow IL-12 production upon TCR stimulation of engineered cells. Although validated in preclinical models, this approach led to significant toxicities in clinical trials, which were associated with promoter leakiness [67]. To address this disappointing result, genetically engineered cells with an activation-induced, membrane-anchored version of IL-12 were generated, limiting diffusion of IL-12 and minimizing off-target effects [68]. Tumor-associated T cells expressing IL-18 upon activation were also proved safe and to induce better tumor clearance in preclinical models [69].

This strategy has been also developed for CAR-T cell models. Multiple cytokine-producing CARs, namely “armored” CAR-T cells (Figure 1), including IL-12, IL-15, and IL-18 [81,82,83] are nowadays at early phases of development. Interestingly, IL-23-secreting CAR-T cells stand out for showing better antitumor efficacy and safety profiles than either, IL-15- and IL-18-secreting CAR-T cells [84]. 

### 4.5. Safety Switches on Engineered CAR-T Cells

As CAR-T cell therapies can induce a large array of severe, life-threatening side effects in patients, tight spatio-temporal control of CAR-T cell viability represents a fundamental objective to make these therapies safer [78,85,86,87,88]. CAR-T cells were engineered to express modified human caspase-9 fused to human FKBP12 (iCasp9 system, Figure 1) such that exogenous administration of AP1930 (small molecule) induces dimerization of FKBP12 and activation of caspase-9, triggering cell apoptosis [70,71]. Another approach to selectively kill CAR-T cells consist of the co-expression of CD20/tEGFR (truncated EGFR), which can be targeted by Rituximab/Cetuximab. Thus, binding of these antibodies to CAR-T cells trigger antibody-dependent cellular cytotoxicity (ADCC) and subsequently killing of engineered cells [89,90,91].

The counterpart of these strategies is irreversibility. Alternatively, more dynamic ON-OFF strategies were developed to temporarily inactivate engineered T cells in cases of severe side effects. To achieve this, TET-ON and TET-OFF systems (Figure 1) have demonstrated to induce/prevent CAR expression on engineered cells by the addition of tetracycline [72,73]. Moreover, inducible promoters responsive to hypoxia-inducible factor (HIF)-1α showed to induce CAR expression only under hypoxic conditions (Figure 1), thus limiting CAR-T cells activation to hypoxic environments, like the tumor cores [74].

Lastly, optogenetics provides a means of tight spatio-temporal control of CAR-T cell activation. Briefly, optogenetics are based on the engineering of target cells with light-inducible sensors which, upon activation, can trigger the expression of actuator proteins in a tight spatio-temporal manner. By using optogenetic devices based on CRY2 (cryptochrome 2)–CIB1 (cryptochrome-interacting basic-helix-loop-helix 1) system, fused to transcription factors involved in target gene expression, scientists successfully induced IL-2, IL-15, and TNFα production [92], as well as CAR expression [75,76] in engineered T cells upon blue-light stimulation, with no relevant leakiness or constitutive activation of the systems, observed so far (Figure 1). The main limitation of this approach is the scarce penetration of visible light into tissues, which might be overcome by using a different kind of waves able to penetrate deeper into tissues (e.g., ultrasounds).

## 5. What’s Next? Future Targets to Increase Long-Term Efficacy of CAR-T Therapy and Unleash T Cells Cytotoxic Potential

We have discussed how bioengineering can aid the design of more effective CAR-T therapies by targeting several shortcomings. Recently, a new constraint has emerged as a limiting factor in CAR-T (and more generally in T cells) long-term action. T cell exhaustion (TCE) is a physiological phenomenon that occurs when T cells are stimulated during long periods of time [93]. Exhausted phenotype is thus frequent in chronic infection and tumor settings. T cell exhaustion is characterized by an exacerbated expression of inhibitory receptors (PD-1, CTLA-4, LAG3, TIM-3) and an impaired proliferation capacity, cytokine production, cytotoxicity as well as a poor memory formation [93]. Gautam S. and colleagues demonstrated that up-regulation of c-Myb, a transcription factor involved in T-cell stemness and memory differentiation, induced the formation of long-lasting T cells a displaying a central memory (TCM) phenotype that increased the antitumor efficacy of T cell therapies in preclinical models [94]. Likewise, the overexpression of c-Jun, another transcription factor involved in T cell stemness, prevented early T cell dysfunction and exhaustion onset and effectively up-regulated genes involved in T-cell memory formation [95].

Other potential targets arose from functional screenings abrogating the expression of candidate genes, as well as transcriptomic analysis of T cells at different maturation stages. By this means, recently published data showed that downregulation of key proteins involved in T cells terminal differentiation and exhaustion such as NR4A, TOX/TOX2, TET2, Regnase1, or PTPN22 [96,97,98,99,100] increased T cell-based treatment half-life and improve anti-tumor performance.

Apart from strategies counteracting T cell exhaustion/favoring T cell stemness, modulation of TCR downstream signaling mediators represents an emerging approach aiming to boost T cell activation and effector functions. CISH, a negative regulator of T cell activation, whose expression is induced upon activating signals such as TCR stimulation or IL-2 engagement, can inhibit T cell activation by inducing PLC-γ1 proteasomal degradation [101]. Seminal studies demonstrated that abrogation of CISH expression on CD8+ T cells improved the anti-tumor efficacy of T cells in mouse models [101]. Scientific literature also points out the role of SOCS1, another member of the SOCS family along with CISH, in preventing T cell overactivation. Thus, down-regulation of SOCS1 also boosted T cells antitumor immune responses in mouse models [102]. From a mechanistic viewpoint, CISH lacks the kinase inhibitory region (KIR), a common motive in other members of the SOCS family. CISH suppresses STAT5 activation by masking STAT5-binding phosphotyrosine motifs on STAT5 cognate receptors (such as EPO, IL-2, IL-3, and IL-5 receptors), and also by inducing ubiquitin/proteasome-dependent degradation of the aforementioned receptors. By contrast, SOCS1 exert a broader inhibitory effect on cytokine signaling since it includes a KIR domain in its protein sequence that is able to inhibit JAK1 and JAK2 tyrosine kinase activity, thus mediating STAT1/STAT3 inhibition. In addition, it has been demonstrated that SOCS1-deficient Tregs easily lose Foxp3 expression and become Th1- or Th17-like effector cells, which can be due to STAT1 and STAT3 hyperactivation [103]. Interestingly, SOCS proteins are also dysregulated in tumor cells, where they usually behave as tumor suppressors. Thus, down-regulation of SOCS proteins is associated with enhanced proliferation and epithelial-to-mesenchymal transition (EMT) for several types of cancer. By contrast, SOCS1 expression in chronic myeloid leukemia correlates with an impaired response to IFN-α and reduced relapsed free survival of those patients [104]. The downregulation of other TCR signaling intracellular mediators such as SHP-1, DGKs, PTPN2286 or RASAL1 [105,106,107,108] also led to better control of tumor growth by genetically-modified T cells in preclinical models. In addition to the TCR downstream modulators described above, CBL-B, an E2 ubiquitin ligase that negatively regulates T cells activation downstream the TCR and CD28 co-stimulatory protein has also being down-regulated in pre-clinical models [21,109,110], where transient down-regulation of this protein resulted in enhanced anti-tumor efficacy and bypassed the requirement for exogenous IL-2 administration for tumor eradication. Phase I clinical trials evaluating the efficacy of CBL-B-knockdown T cells are currently being conducted (NCT03087591).

## 6. Conclusions

Synthetic biology has demonstrated as a powerful engine in the scenario of cancer immunotherapy, especially with the advent of CAR-T cells therapy in recent years. However, several efforts are still needed to improve the design of genetic devices with long-term stability and efficacy. It has been recently shown that exogenous payloads can impact cellular physiology by using intracellular resources [111]. This was demonstrated to in turn affect circuit performances unless specific network designs are implemented to mitigate this phenomenon [111]. Moreover, safety concerns of engineered immune cells increase the awareness that further engineering of devices with higher efficacy, modularity, and tunability is required. Although most of the strategies applied to date address the primary recognition of a TAA and T cell activation, advances in the development of strategies to control the TME taking use of controllable and complex genetic circuits can indeed boost T and CAR-T cell therapy efficacy [112]. To this end studies that take advantage of 3D scaffolds to mimic the tumor microenvironment [113] and that can modulate cell functionality by fine-tuning cell-material crosstalk [114] may shed light on new parameters for innovative bioengineering designs. 

In parallel, modularity-based strategies could substantially reduce the costs and time of T cell expansion into antigen-specific CAR-T cells and therefore significantly accelerate the development of new approaches applied to the clinic. As the biology of T cells is complex as any other mammalian cells, circuit design should be precisely adjusted to prevent or revert non-expected behavior as it is observed within some cells becoming exhausted. To engineer the next generation of CAR-T cell therapy and increase long-term efficacy, associated sensor-actuator strategies should be considered to promote T cell stemness and prevent dysfunction. The increasing knowledge in T cell biology associated with the modular and controllable approaches will allow us to develop innovative synthetic immunology therapeutic strategies to fight cancer and other immune diseases.

## Figures and Tables

**Figure 1 vaccines-08-00732-f001:**
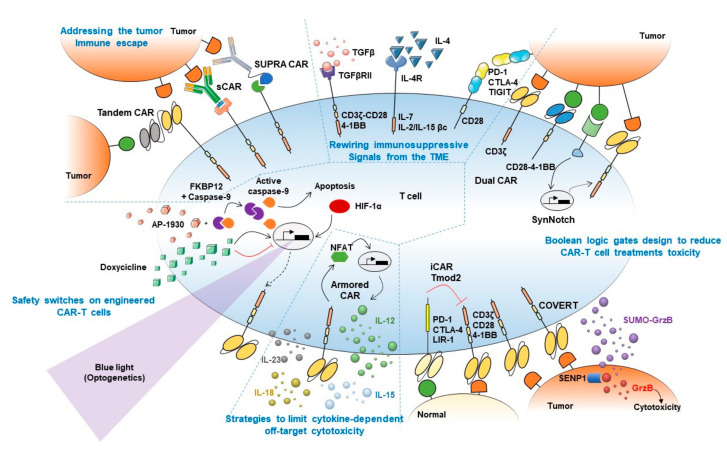
Synthetic biology approaches to boost CAR-T cell treatment’s efficacy. Graphical abstract summarizing main synthetic biology approaches developed to prevent current limitations of adoptive T cell therapies. Strategies to boost CAR-T cell therapy efficacy are mechanistically classified into (1) approaches to overcome tumor immune escape (top left), (2) strategies to rewire immunosuppressive signals from the tumor microenvironment (TME) (top middle), (3) Boolean logic gates to reduce CAR-T cell treatment toxicity (right), (4) strategies to limit cytokine-dependent off-target cytotoxicity (bottom middle), and (5) safety switches on engineered CAR-T cells (bottom left). Main strategies to overcome tumor antigen escape depicted in the figure include Tandem (or bi-specific) CAR, switchable CAR or sCAR, and split, universal, and programmable CAR or SUPRA CAR. Fusion proteins combining TGFβ/IL-4 immunosuppressive ectodomains with CD3/CD28-4-1BB or IL-7/IL-2-IL-15 endodomains are included among approaches devised to rewire immunosuppressive signal from the TME. Approaches focused on reducing CAR-T cell treatment toxicity depicted in the figure include Dual CAR, SynNotch, and COVERT strategies, which depend on the presence in target tumor cells of two different antigens to allow T cells activation, and iCAR/Tmod2 approaches that prevent activation of T cells against normal cells that express the antigen recognized by the inhibitory CAR. Strategies to limit cytokine-dependent off-target cytotoxicity include Armored CARs, which can secrete cytokines locally upon activation, preventing systemic undesired side effects induced by repeated systemic cytokines administration. Safety switches included in the figure comprise the iCasp system, which triggers engineered cells apoptosis upon AP-1930 administration, as well as TET-inducible, hypoxia-inducible and light-inducible CAR expression to allow tight control of CAR expression upon doxycycline treatment, hypoxia onset, or blue light stimulation, respectively.

**Figure 2 vaccines-08-00732-f002:**
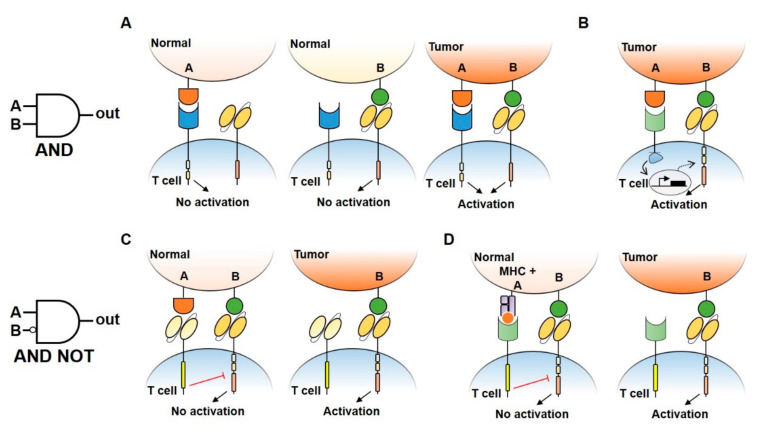
Logic gates design to reduce CAR-T cell treatment toxicity. (**A**) An AND gate CAR-T design consisting of splitted intracellular activation (CD3ζ signaling domain in the CAR receptor and CD28/4–1BB costimulatory domains in another receptor). Single antigen recognition is not able to activate T cell response, preventing damage in normal tissues. Double antigen recognition (A AND B) drives the activation of T cells and killing of the tumor. (**B**) An AND gate approach using a synNotch receptor combined with a CAR receptor to increase antigen specificity and safety. Under the recognition of a first antigen by synNotch receptor, an intracellular transcription factor (TF) is released and induces the activation of CAR receptor specific to a second antigen, allowing the activation of T cells. (**C**,**D**) By using inhibitory CAR receptors (iCAR) (**C**) or blocking Tmod2 (**D**), AND NOT gates allow the engineered cells to distinguish normal tissues by preventing T cell activation when a double antigen recognition is present.

**Table 1 vaccines-08-00732-t001:** Synthetic biology approaches to boost cancer immunotherapy

Synthetic Biology Approach	Control Module	Purpose	References
Tandem CAR	Logic Gate OR	Avoid OFF-target	[45,46,47]
sCAR	Logic Gate AND	Increase specificity	[48,49,50,51]
SUPRA CAR	Docking system	Tune T cell activation	[52]
Engineered TGF-β-RII	Chimeric receptors/Rewiring	Overcome TME immunosuppression	[53,54]
Engineered IL-4-R	Chimeric receptors/Rewiring	Overcome TME immunosuppression	[55,56]
Engineered IRs	Chimeric receptors/Rewiring	Overcome TME immunosuppression	[57,58]
CCR ^a^-CAR	Logic Gate AND	Increase specificity	[59]
SynNotch CAR	Logic Gate AND	Increase specificity	[60]
COVER-T	Logic Gate AND-like	Avoid OFF-target	[61]
iCAR	Logic Gate AND NOT	Avoid OFF-target	[62]
Tmod2	Logic Gate AND NOT	Target LOH	[63]
Engineered IL-2R/IL-7R	Cytokine signaling	Reduce cytokine cytotoxicity	[64,65]
Engineered IL-12/IL-18	Cytokine signaling	Reduce cytokine cytotoxicity	[66,67,68,69]
iCasp9	Safety switches	Control CAR-T cell viability	[70,71]
TET-ON/TET-OFF	Safety switches	Reduce side effects	[72,73]
HIF-CAR	Safety switches	Activate T cell only in the tumor core	[74]
Blue light-activated CAR	Safety switches	Tune T cell activation	[75,76]

^a^ CCR = Chimeric costimulatory receptor.

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
