# Peer review of "Precision Tools in Immuno-Oncology: Synthetic Gene Circuits for Cancer Immunotherapy"

_vaccines, 2020, doi:10.3390/vaccines8040732_

Round 1

Reviewer 1 Report

The manuscript "Precision tools in immuno-oncology: Synthetic gene 2 circuits for cancer immunotherapy" show a good review in Immuno approaches strategy against cancer.

The authors summarized an elegant view of Synthetic biology and discussed well the pros and cons of the therapeutic tools, especially with the advent of CAR-T cell therapy in recent years.

The figures have high quality and at the same time explain the main points of how to work immunotherapy.

I think that the manuscript is suitable to be published in "Vaccine"

Author Response

Reviewer 1:
Comments and Suggestions for Authors
The manuscript "Precision tools in immuno-oncology: Synthetic gene 2 circuits for cancer immunotherapy" show a good review in Immuno approaches strategy against cancer.
The authors summarized an elegant view of Synthetic biology and discussed well the pros and cons of the therapeutic tools, especially with the advent of CAR-T cell therapy in recent years.
The figures have high quality and at the same time explain the main points of how to work immunotherapy.
I think that the manuscript is suitable to be published in "Vaccine"
Answer: We thank the reviewer for the great appreciation of our manuscript.

Reviewer 2 Report

The authors present an overview of recent synthetic biology strategies for mammalian cell therapies with relevance to the next generation of genetic circuits for cancer immunotherapy. The authors indicate that CAR-T therapy represent the state of the art of cell-based therapies and CAR-T cell design have improved over time leading to several generations that are distinguished for changes in the co-immunostimulatory domains. The authors report that CAR-T cell therapies suffer limitations and novel bioengineering approaches have been implemented to help to overcome them and discuss how bioengineering can aid the design of more effective CAR-T therapies by targeting several shortcomings. CAR-T cells therapies induce a large array of severe, life-threatening side effects in patients and the review indicates that tight spatio-temporal control of CAR-T cells viability represents a critical approach to make a safer therapy. The authors conclude that safety concerns of engineered immune cells (TAA and T cell) have stimulated the need for further engineering of devices with higher efficacy, modularity and tunability. Advances in the development strategies to control the tumour microenvironment with the use novel genetic circuits will determine T and CAR-T cell therapy efficacy with relevance to cancer immunotherapy.

Comments:

  1. The literature shows that the CAR T-cell therapy success rate is about 30% to 40% for lasting remission. Can the authors provide an estimation of the implementation of the novel bioengineering approaches and their success rate (eg 70-90%) in patients?
  2. The use of CAR-T cell therapies can have serious side effects. The authors may expand on these side effects that include neurotoxicity or changes in the brain that cause swelling, confusion, seizures, or severe headaches. High fevers and dangerously low blood pressure within days after CAR T-cell therapy and the toxic effects of the cytokine release syndrome.
  3. The genetic circuits that determine the novel CAR-T therapies may have effects on the primary and secondary immune response. Can the authors provide evidence to indicate that the novel CAR-T therapy do not have toxic effects on B cells relevant to bacterial contamination?
  4. Do the CAR-T cell therapy need to be used when the cell anti-aging genes are activated such as Sirtuin 1 critical for survival of cells? The literature reports that CAR-T cell therapy can result in adverse events in any organ system. Sirtuin 1 is involved with the immune system and multiple organ disease syndrome. Is there a difference in CAR-T therapy with relevance to Sirtuin 1 repression versus activation with relevance to cancer immunotherapy?

REFERENCES:

  1. Zhang C, Durer S, Thandra KC, et al. Chimeric Antigen Receptor T-Cell Therapy. [Updated 2020 Oct 11]. In: StatPearls [Internet]. Treasure Island (FL): StatPearls Publishing; 2020 Jan.
  2. Anti-Aging Genes Improve Appetite Regulation and Reverse Cell Senescence and Apoptosis in Global Populations. Advances in Aging Research, 2016, 5, 9-26
  3. Single Gene Inactivation with Implications to Diabetes and Multiple Organ Dysfunction Syndrome. J Clin Epigenet. 2017. Vol. 3 No. 3:24.

Author Response

 Reviewer 2:

Comments and Suggestions for Authors

The authors present an overview of recent synthetic biology strategies for mammalian cell therapies with relevance to the next generation of genetic circuits for cancer immunotherapy. The authors indicate that CAR-T therapy represent the state of the art of cell-based therapies and CAR-T cell design have improved over time leading to several generations that are distinguished for changes in the co-immunostimulatory domains. The authors report that CAR-T cell therapies suffer limitations and novel bioengineering approaches have been implemented to help to overcome them and discuss how bioengineering can aid the design of more effective CAR-T therapies by targeting several shortcomings. CAR-T cells therapies induce a large array of severe, life-threatening side effects in patients and the review indicates that tight spatio-temporal control of CAR-T cells viability represents a critical approach to make a safer therapy. The authors conclude that safety concerns of engineered immune cells (TAA and T cell) have stimulated the need for further engineering of devices with higher efficacy, modularity and tunability. Advances in the development strategies to control the tumour microenvironment with the use novel genetic circuits will determine T and CAR-T cell therapy efficacy with relevance to cancer immunotherapy.

Answer: We thank the reviewer for the valuable comments provided, that we address below. Based on the comments, we believe we now have a new improved version of the manuscript.

Comments:

  1. The literature shows that the CAR T-cell therapy success rate is about 30% to 40% for lasting remission. Can the authors provide an estimation of the implementation of the novel bioengineering approaches and their success rate (eg 70-90%) in patients?

Answer: Thanks for the interesting question. Given the vast number of approaches already in pre-clinical phases, we are positive that in the next few years the success rate will increase. At the same time though, until clinical trials will start, we believe it is hard to give a quantitative estimation of the success rate and we would rather not overreach ourselves.

  1. The use of CAR-T cell therapies can have serious side effects. The authors may expand on these side effects that include neurotoxicity or changes in the brain that cause swelling, confusion, seizures, or severe headaches. High fevers and dangerously low blood pressure within days after CAR T-cell therapy and the toxic effects of the cytokine release syndrome.

Answer: We thank the reviewer for the insightful comment. Indeed there is unfortunately still a number of side effects, some of which discussed as issue to address by bioengineering approaches. Indeed neurotoxicity is still a major concern. We have now included a further discussion on these side effects as consequence of the use of CAR T cell therapies in the revised manuscript, lines 118-129.

  1. The genetic circuits that determine the novel CAR-T therapies may have effects on the primary and secondary immune response. Can the authors provide evidence to indicate that the novel CAR-T therapy do not have toxic effects on B cells relevant to bacterial contamination?

Answer: We thank the review for rinsing this discussion. Indeed the integration of synthetic circuits in mammalian cells (including also T cells) can lead to immune response or epigenetic silencing when foreign components (e.g. bacterial genetic modules are used). In the specific case of the latest CAR-T cell engineering for human cancer therapy, based on the genome integration of genetic parts by lentivirus infection, no direct bacterial or other foreign components are used. In addition, during manufacturing a series of quality control and nonclinical checks are performed to avoid, among other things, bacterial contamination (Li et al., 2019).

References

Li, Y., Huo, Y., Yu, L., and Wang, J. (2019). Quality Control and Nonclinical Research on CAR-T Cell Products: General Principles and Key Issues. Engineering 5, 122–131.

  1. Do the CAR-T cell therapy need to be used when the cell anti-aging genes are activated such as Sirtuin 1 critical for survival of cells? The literature reports that CAR-T cell therapy can result in adverse events in any organ system. Sirtuin 1 is involved with the immune system and multiple organ disease syndrome. Is there a difference in CAR-T therapy with relevance to Sirtuin 1 repression versus activation with relevance to cancer immunotherapy?

REFERENCES:

  1. Zhang C, Durer S, Thandra KC, et al. Chimeric Antigen Receptor T-Cell Therapy. [Updated 2020 Oct 11]. In: StatPearls [Internet]. Treasure Island (FL): StatPearls Publishing; 2020 Jan.
  2. Anti-Aging Genes Improve Appetite Regulation and Reverse Cell Senescence and Apoptosis in Global Populations. Advances in Aging Research, 2016, 5, 9-26
  3. Single Gene Inactivation with Implications to Diabetes and Multiple Organ Dysfunction Syndrome. J Clin Epigenet. 2017. Vol. 3 No. 3:24.

Answer: We thank we reviewer for this interesting question. The effects of Sirt1 dysregulation depend on the cell subset we consider (tumor cells, T cells or whole body causing premature aging or metabolic syndrome). For example, it has been demonstrated that Sirt1 overexpression on tumor cells induces resistance to classical chemotherapies (Cao et al., 2015; Vellinga et al., 2015). Although the relationship between Sirt1 expression in tumors and immunotherapy outcome in treated patients has not being deeply studied so far, we can hypothesize that tumor cells up-regulating Sirt1 might be more resistant to immunotherapy compared with tumor cells expressing lower levels of Sirt1, since Sirt1 expression is inversely correlated with tissue inflammation. In the same way, patients showing a global down-regulation of Sirt1 caused by metabolic syndrome or other age-related syndromes, show aberrantly prominent levels of inflammation in affected tissues. Consequently, a higher incidence of off-target toxicity (CRS, neurotoxicity) is expected in these patients, since a predominantly pro-inflammatory microenvironment might facilitate T cell activation in inflamed tissues (Martins, 2016, 2017). Concerning T cell biology, Sirt1 inhibition can increase Foxp3 acetylation and promote the production and functions of Foxp3+ T-regulatory (Treg) cells, whereas the acetylation of RORγt decreases its transcriptional activity DNA binding and decreases the differentiation of proinflammatory Th17 cells (Chadha et al., 2019). Thus, Sirt1 downregulation can impair T cell fitness to fight cancer, since Treg cells are known to be involved in resistance to adoptive cell therapy. In addition, the cellular energetic pathway is also a key feature of the unresponsive state of anti-tumor T cells. Recently, it was demonstrated that CD38-NAD+- Sirt1 axis acts as a key determinant of the therapeutic efficacy of anti-tumor T cells. T cells with genetic ablation of CD38 not only exhibited complete remission of tumor growth but also improved the durability of the response. Most, notable feature observed in T cells with CD38 deficiency was its preferential reliance on glutaminolysis to empower OXPHOS. In addition to facilitate glutaminolysis, CD38 deficiency in T cells resulted in elevated intracellular level of nicotinamide adenosine dinucleotide (NAD+), a metabolite that acts as a substrate for Sirtuin (Sirt) family of enzymes catalyzing deacetylation. Thus, NAD+-Sirt1 axis, which inversely correlates with CD38 expression, plays a role in epigenetically regulating various transcription factors/molecules that endow T cells with stemness (FOXO1, EOMES) and superior anti-tumor response (GRZB) (Chatterjee et al., 2019).

We have not included this part in the discussion of the revised manuscript for length limitations. We leave to the discretion of the Editor whether to include it exceeding the size limit.

References

Cao, B., Shi, Q., and Wang, W. (2015). Higher expression of SIRT1 induced resistance of esophageal squamous cell carcinoma cells to cisplatin. J. Thorac. Dis. 7, 711–719.

Chadha, S., Wang, L., Hancock, W.W., and Beier, U.H. (2019). Sirtuin‐ 1 in immunotherapy: A Janus‐ headed target. J. Leukoc. Biol. 106, 337–343.

Chatterjee, S., Chakraborty, P., and Mehrotra, S. (2019). CD38-NAD+-Sirt1 axis in T cell immunotherapy. Aging (Albany. NY). 11, 8743–8744.

Li, Y., Huo, Y., Yu, L., and Wang, J. (2019). Quality Control and Nonclinical Research on CAR-T Cell Products: General Principles and Key Issues. Engineering 5, 122–131.

Martins, I.J. (2016). Anti-Aging Genes Improve Appetite Regulation and Reverse Cell Senescence and Apoptosis in Global Populations. Adv. Aging Res. 05, 9–26.

Martins, I.J. (2017). Single Gene Inactivation with Implications to Diabetes and Multiple Organ Dysfunction Syndrome. J. Clin. Epigenetics 03.

Vellinga, T.T., Borovski, T., de Boer, V.C.J., Fatrai, S., van Schelven, S., Trumpi, K., Verheem, A., Snoeren, N., Emmink, B.L., Koster, J., et al. (2015). SIRT1/PGC1 - Dependent Increase in Oxidative Phosphorylation Supports Chemotherapy Resistance of Colon Cancer. Clin. Cancer Res. 21, 2870–2879.

Reviewer 3 Report

This review paper summarized the present state of knowledge on the organizing cancer immunotherapy and synthetic biology strategies for mammalian cell therapies, especially CAR T-cell therapy in cancer. Moreover, authors discuss the current mechanism of CAR-T cell therapy. Finally, authors point out the current genetic engineering of immunotherapy included T-cell proliferation or half life in designed needs to be improved. And through the recent publication, they organizing how to make CAR-T therapies more efficient by bioengineering.

Specific comments

  1. Please describe more detailed information about the latest cancer vaccines.
  2. In figure 1, the classification needs to be clarified and please describe detail in text.
  3. The title of this article states that it should be a review of all immunotherapies, but most of the article has sorted out the CAR-T cell technology. In the abstract part, it should be stated that this article is about importance of CAR-T cell genetic engineering.
  4. line 70 and 71, there are two types of ACT cell but no explanation what is tumor-infiltrating lymphocytes(TILs) from an endogenous source.
  5. line 118, the “AND” abbreviations should be defined.
  6. Please describe why and how optogenetics technology control CAR-T cell.
  7. In the conclusion part, need to compare the therapy about the downregulation of key protein CISH expression/SOCI function both in solid tumor and blood cancer.

Minor comments

  1. Please provide a list of abbreviations in the supplemental material.
  2. Please corrected chapter label “conclusion part” from 5 to 6.

Author Response

 Reviewer 3:

Comments and Suggestions for Authors

This review paper summarized the present state of knowledge on the organizing cancer immunotherapy and synthetic biology strategies for mammalian cell therapies, especially CAR T-cell therapy in cancer. Moreover, authors discuss the current mechanism of CAR-T cell therapy. Finally, authors point out the current genetic engineering of immunotherapy included T-cell proliferation or half life in designed needs to be improved. And through the recent publication, they organizing how to make CAR-T therapies more efficient by bioengineering.

Answer: We would like to thank the reviewer for the insightful comments which we address below.

Specific comments

  1. Please describe more detailed information about the latest cancer vaccines.

Answer: We thank the reviewer for making this observation. We now provide a more detailed explanation about therapeutic and prophylactic cancer vaccines, with a focus on the synthetic biology approach, lines 56-68.

  1. In figure 1, the classification needs to be clarified and please describe detail in text.

Answer: We thank the reviewer for pointing that the picture need more clarification. A more detailed version of figure 1 is now provided in the revised version of the manuscript, at lines 230-249.

  1. The title of this article states that it should be a review of all immunotherapies, but most of the article has sorted out the CAR-T cell technology. In the abstract part, it should be stated that this article is about importance of CAR-T cell genetic engineering.

Answer: We thank the reviewer for the valid observation. Indeed this special issue of Vaccines relates to different types of immunotherapies, and in our review we decided to focus on the precision tools that bioengineering can offer to ensure better cell-based immunotherapies. As the state-of-the-art approach in cell-based immunotherapy is CAR-T cell therapy, we sought to discuss more the intertwining of synthetic biology tools and CAR-T cell therapy improvements. We included the suggestion of the reviewer about the focus on the CAR-T cell genetic engineering importance in the revised new version of the manuscript, lines 18-19.

  1. line 70 and 71, there are two types of ACT cell but no explanation what is tumor-infiltrating lymphocytes(TILs) from an endogenous source.

Answer: We thank the reviewer for pointing the absence of TILs explanation. TILs therapy is now included in the new version of the manuscript, lines 82-87.

  1. line 118, the “AND” abbreviations should be defined.

Answer: We thank the reviewer for asking more clarification about this topic. The “AND” refers to a Boolean logic gate type, rather than being an acronym. The AND gate indicate that two inputs must be present in order to activate the circuit (and thus the genetic response). On the contrary, the presence of either of a single input does not result in the desired output. For better understanding, we included it in quotation marks and added the word “logic” before “gate”, lines 145-153 of the revised new version of the manuscript.

  1. Please describe why and how optogenetics technology control CAR-T cell.

Answer: We thank the reviewer for pointing out that this part requires a better description. A more detailed explanation about how optogenetics work in general and in CAR-T cell therapy is now provided in the revised version of the manuscript, lines 365-370.

  1. In the conclusion part, need to compare the therapy about the downregulation of key protein CISH expression/SOCI function both in solid tumor and blood cancer.

Answer: We thank the reviewer for the comment. Mechanistic differences between CISH and SOCS1 mechanisms of action have been further detailed in the manuscript. In addition, a brief description of the role of these two SOCS proteins in tumor biology has been added to the main text, lines 402-416. Since we didn’t find any paper comparing the effectiveness of CISH and SOCS1 KO T cell adoptive immunotherapies in the treatment of cancer we do not know which therapy might be more effective to eradicate either solid and liquid cancers in patients.

Minor comments

  1. Please provide a list of abbreviations in the supplemental material.

Answer: We thank the reviewer for rising this need. We provided in the improved version of this manuscript a list of abbreviations suggested, that will be included in the supplemental material.

    2. Please corrected chapter label “conclusion part” from 5 to 6.

Answer: We thank the reviewer for making this observation. We now corrected as requested.

Round 2

Reviewer 2 Report

The authors have ensured that the research is properly verified and have revised the manuscript according to the reveiwer’s comments. The revised manuscript maintains the high standards for peer-reveiwed journals.